# TEACHER THAT LISTENS: ADAPTIVE STUDENT-AWARE DISTILLATION FOR REASONING

## ABSTRACT

Knowledge distillation is a standard approach to compress the capabilities of large language models into smaller students. However, standard distillation methods often produce suboptimal results due to a mismatch between teacher-generated rationales and the student's specific learning requirements. In this paper, we introduce the Adaptive student-aware Distillation for Reasoning (AdaptDistill), designed to bridge this gap by iteratively identifying the student's errors and allowing the teacher to refine its explanations according to the student's needs. Each iteration directly targets the student's learning deficiencies, motivating the teacher to provide tailored rationales that specifically address these weaknesses for better learning. Empirical evaluations on various challenging mathematical and commonsense reasoning tasks demonstrate that our adaptive distillation approach, AdaptDistill, significantly outperforms standard distillation methods, achieving significant performance gains. Our work fundamentally reframes knowledge distillation as an iterative *teacher–student* interaction, effectively leveraging dynamic refinement by the teacher for better knowledge distillation.

## 1 INTRODUCTION

An increasingly popular approach for reducing the size and computational demands of large language models (LLMs) involves distilling their knowledge into compact, high-performing student models (Hinton et al., 2015; Sanh et al., 2020; Zhu et al., 2024). Initially, the primary approach involved aligning the student's outputs with the soft logits of the teacher, enabling the student to approximate the teacher's distribution (Hinton et al., 2015). However, recently, this paradigm has expanded to include reasoning chains, which seek to instill smaller models with interpretable intermediate steps rather than simply matching final outputs (Shridhar et al., 2023; Hsieh et al., 2023; Li et al., 2023; Chenglin et al., 2024; Liu et al., 2024).

While effective, these approaches still rely on a one-shot distillation process,[1] where the teacher produces rationales independently of the student's behavior. This leads to a distributional mismatch: the teacher's rationales often reflect its own reasoning style rather than addressing the student's concrete mistakes or learning bottlenecks (Agarwal et al., 2024; Adarsh et al., 2025). As a result, the student may fail to fully benefit from teacher supervision, limiting both reasoning fidelity and transferability.

We propose Adaptive student-aware Distillation for Reasoning, an *iterative*, adaptive distillation framework that closes the loop between teacher and student. In each round, the teacher generates rationales to teach the given task to the student; the student learns it and attempts the task that exposes its learning gaps. The teacher then *optimizes its rationale* by conditioning on a scored history based on whether the student can solve the task correctly or not along with the learning gaps and produces a refined, *targeted* explanation. The student is then fine-tuned on a curated mix of its *own* correct traces (to preserve successful behaviors) and the teacher's corrected ones (to address gaps). Repeating this process personalizes supervision to the student rather than reflecting generic teacher behavior.

---

[1] We refer to training a smaller model on a larger model's outputs as *one-shot distillation*, i.e., without any iterative student–teacher interaction.

This loop mimics the optimization process in the context window of the teacher as every iteration maintains a trajectory of (rationales, score) pairs and proposes the next candidate rationale to improve an *objective* of improving the student accuracy. In this sense, AdaptDistill instantiates a derivative-free optimization loop driven by an LLM that reasons over its own scored attempts and the student's responses to iteratively improve guidance. An overview of the loop is shown in Figure 1.

We evaluate AdaptDistill on challenging mathematical reasoning benchmarks, including MATH (Lightman et al., 2024) and MMLU Pro (Wang et al., 2024). Empirically, our iterative strategy consistently outperforms one-shot distillation, with accuracy improvements of up to $20\%$. We further find that the teacher-generated datasets produced by AdaptDistill retain their effectiveness across diverse student architectures, and that students trained with our approach maintain (and sometimes improve) out-of-domain performance. These results suggest that our framework improves not only in-distribution accuracy but also the transferability of reasoning skills.

In summary, our work introduces a shift in perspective: distillation is no longer a one-time transfer of knowledge, but an ongoing adaptive teacher–student interaction that produces more robust and generalizable student models.

## 2 RELATED WORK

**Knowledge distillation**  Extensive research has explored knowledge distillation since (Hinton et al., 2015) introduced the concept of training smaller student models using soft-label outputs (soft logits) from larger teacher models. Early influential models such as DistilBERT (Sanh et al., 2020) demonstrated substantial parameter reductions ( 40%) without significant performance losses. Subsequent advancements, like MiniLM (Wang et al., 2020), further refined this technique by focusing specifically on distilling the self-attention mechanism. More recent works, including (Shum et al., 2024) and (Timiryasov & Tastet, 2023), explored selective training strategies by choosing essential samples from the teacher's training set.

The integration of Chain-of-Thought (CoT) reasoning (Wei et al., 2022) into distillation has sparked additional innovations. For example, (Shridhar et al., 2023) trained student models to explicitly replicate the teacher's CoT reasoning steps. Similarly, (Li et al., 2023) generated diverse rationales for individual questions, while (Chenglin et al., 2024) extended this by incorporating multiple reasoning styles. (Zhu et al., 2024) distilled mathematical reasoning into structured equations and combined various reasoning modalities (text, code, equations). Further, (Luo et al., 2025) leveraged reinforcement learning to fine-tune student models. Despite these advancements, most approaches remain one-directional, with teacher outputs being independent of student-specific needs. In contrast, our method explicitly addresses the student's learning gaps by encouraging the teacher to adapt their rationales based on the student needs.

**Iterative distillation**  Our work draws significant inspiration from iterative forms of knowledge distillation. Previous research often emphasized the transfer of intermediate reasoning steps to smaller models, either sequentially (Hsieh et al., 2023; Magister et al., 2023; Shridhar et al., 2023) or concurrently integrating various reasoning strategies (Chenglin et al., 2024; Zhu et al., 2024; Jain et al., 2024). Notably, (Wang et al., 2023) employed iterative interactions between students and black-box teachers, with teachers providing targeted feedback based on student failures. Similarly, (Adarsh et al., 2025) combined iterative self-guided training and self-generated data to enhance reasoning in smaller models. Additionally, (Agarwal et al., 2024) utilized on-policy training to better align student and teacher distributions by incorporating student-generated outputs. Self-distillation methods (Zelikman et al., 2022; Liu et al., 2024) involved models generating, filtering, and fine-tuning based on their own rationales. Reinforcement learning-based self-training was further explored by (Gulcehre et al., 2023), where off-policy sampling informed dataset generation. Unlike these methods, our iterative approach uniquely emphasizes a dynamic, adaptive feedback loop where the teacher explicitly targets and rectifies the student's learning challenges.

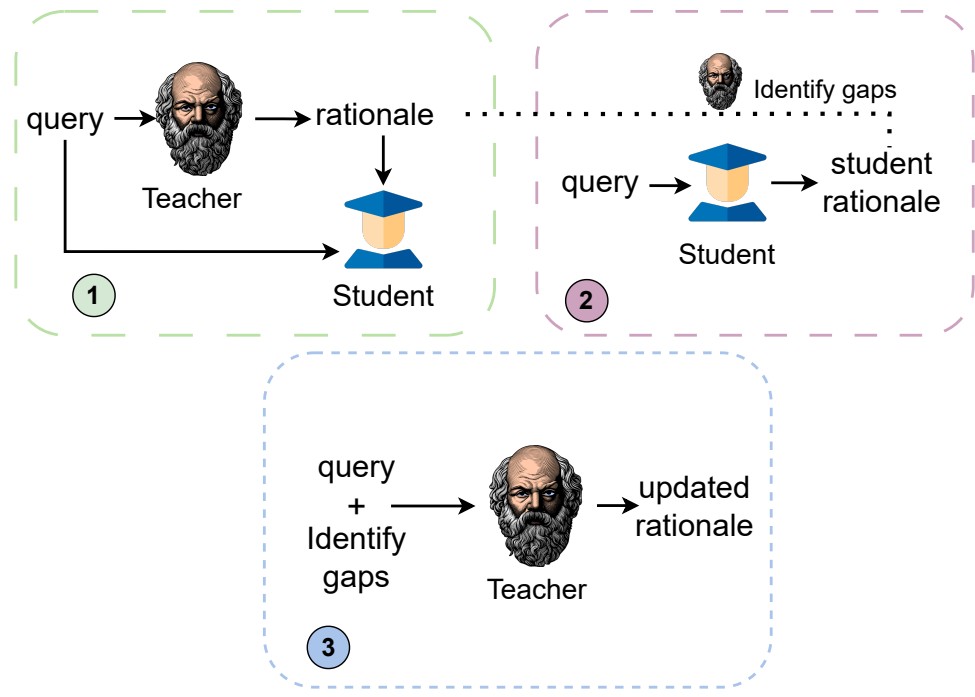

Figure 1: **Overview of AdaptDistill.** (1) The teacher generates an initial rationale for a query and supervises the student. (2) The student attempts the same query; the teacher compares the student's rationale with its own prior rationale to identify *learning gaps*. (3) Re-prompted with the query plus gap information (and a scored history on the validation set), the teacher produces an *updated rationale* targeted to the student's weaknesses. Steps 1–3 repeat: the student is fine-tuned on curated traces while the teacher keeps track of progress in its prompt, effectively acting as an in-context optimizer of rationales.

## 3 METHODOLOGY

In this section, we first outline the standard knowledge distillation process and discuss its key limitations. We then introduce our proposed Adaptive student-aware Distillation for Reasoning framework, which addresses these limitations by enabling the teacher to iteratively identify the student's weaknesses and refine the rationale being distilled accordingly.

**Problem Setup.** We consider a large teacher model $p_L$ (LLM) and a smaller student model $p_{sm}^{\theta}$ (with learnable parameters $\theta$). We have a dataset of math questions $\mathcal{D} = \{(q_i, a_i)\}_{i=1}^{n}$, and an associated held-out validation set $\mathcal{V} = \{(q_j, a_j)\}_{j=1}^{m}$ to monitor progress at each iteration. Our goal is to distill the teacher's reasoning capabilities into the student iteratively by constantly moderating the errors made by the student and targeted refinements from the teacher. We use a fixed instruction $I$ to elicit chain-of-thought (CoT) from both teacher and student.

### 3.1 STANDARD KNOWLEDGE DISTILLATION

We follow (Shridhar et al., 2023; Magister et al., 2023). The teacher $p_L(r \mid q, I)$ is prompted to produce a rationale (chain-of-thought) $r_i$ for each question $q_i$. Let $\widehat{a}_i$ be the final numeric result extracted from $r_i$ (denoted $\text{ans}(r_i)$). We keep only those samples for which $\widehat{a}_i = a_i$, yielding a filtered dataset $\mathcal{D}_{\text{LLM}} = \{(q_i, r_i)\}_{i=1}^{N}$ of correct (question, rationale) pairs.

The student is then fine-tuned by maximizing the log-likelihood of these rationales:

$$\mathcal{L}_{\mathrm{L}}(\theta) \;=\; -\frac{1}{|\mathcal{D}_{\mathrm{LLM}}|} \sum_{(q_i, r_i) \in \mathcal{D}_{\mathrm{LLM}}} \sum_{t=1}^{M_i} \log p_{\mathrm{sm}}^{\theta}\big(r_{i,t} \,\big|\, r_{i,<t}, q_i, I\big), \tag{1}$$

where $M_i$ is the number of tokens in $r_i$. Minimizing $\mathcal{L}_{\mathrm{L}}(\theta)$ via gradient descent adjusts $\theta$ so that the student mimics the teacher's rationales. This is the baseline *one-step* standard knowledge distillation.

## 3.2 ADAPTIVE STUDENT-AWARE DISTILLATION FOR REASONING

Standard Knowledge Distillation can result in a discrepancy between the output distributions of the teacher and student models, which may hinder effective learning (Agarwal et al., 2024; Adarsh et al., 2025). To counter this, we propose an *iterative* distillation strategy AdaptDistill that repeatedly identifies the student's errors and lets the teacher produce improved rationales that focus on the *learning gaps* to improve the student's performance. The setup is visualized in Figure 1.

We perform $K$ such iterations, maintaining updated student parameters $\theta^{(k)}$ after each round. Each step is explained below.

**1) Identify Student Learning Gaps & Incorporate in Teacher Prompt.** At each iteration $k \in \{1, \dots, K\}$, the student $p_{\mathrm{sm}}^{\theta^{(k-1)}}(r \mid q, I)$ answers each problem in the validation set $\mathcal{V}$. The student's generation for $q_j \in \mathcal{V}$ is denoted $\tilde{r}_j^{(k,\mathrm{student})}$. We compare the final numeric result $\mathrm{ans}\big(\tilde{r}_j^{(k,\mathrm{student})}\big)$ to $a_j$ and obtain a binary score $\mathrm{score}_j^{(k)} \in \{0, 1\}$. We collect these labeled pairs and maintain a running history across iterations:

$$\mathcal{H}^{(k)} \;:=\; \bigcup_{\ell=1}^{k} \Big\{ \big(q_j, \; \tilde{r}_j^{(\ell,\mathrm{student})}, \; \mathrm{score}_j^{(\ell)} \big) \Big\}_{j=1}^{m}.$$

This history is shown to the teacher in its prompt context so the teacher can observe the student's progress and mistakes.

**2) Regenerate from the Teacher to adapt to the student needs.** We now consider the *training* dataset $\mathcal{D}$ to improve the student. For each $(q_i, a_i) \in \mathcal{D}$, retrieve: (i) the student's most recent generation for $q_i$,

$$\tilde{r}_i^{(\le k-1,\mathrm{student})} \;:=\; \begin{cases} \tilde{r}_i^{(k-1,\mathrm{student})}, & \text{if available,} \\ \tilde{r}_i^{(\mathrm{init})}, & \text{otherwise,} \end{cases}$$

and (ii) the teacher's previous rationale $r_i^{(k-1,\mathrm{teacher})}$ (if it exists). Define the indicator of the student being already correct:

$$S_i^{(k-1)} \;:=\; \mathbf{1}\Big\{ \mathrm{ans}\big(\tilde{r}_i^{(\le k-1,\mathrm{student})}\big) = a_i \Big\}.$$

**Use-student rule.** If $S_i^{(k-1)} = 1$ (the student is already correct), we *do not* query the teacher; instead, we add the student's own rationale $\tilde{r}_i^{(\le k-1,\mathrm{student})}$ to the training set for this question. This was done following (Adarsh et al., 2025) and (Agarwal et al., 2024) as adding student's own generations helps in reducing the distributional mismatch between the teacher and the student.

Otherwise (i.e., if $S_i^{(k-1)} = 0$), we construct gap information

$$\Delta_i^{(k)} \;:=\; \big( \tilde{r}_i^{(\le k-1,\mathrm{student})}, \; r_i^{(k-1,\mathrm{teacher})}, \; \mathcal{H}^{(k)} \big),$$

and sample a new teacher rationale

$$r_i^{(k,\mathrm{teacher})} \;\sim\; p_L\big(\cdot \mid q_i, I, \Delta_i^{(k)}\big).$$

Let $C_i^{(k)} := \mathbf{1}\{\mathrm{ans}(r_i^{(k,\mathrm{teacher})}) = a_i\}$ denote the acceptance event.

The curated dataset for iteration $k$ then aggregates correct student traces and (filtered) teacher traces:

$$\mathcal{D}_{\mathrm{LLM}}^{(k)} \;:=\; \underbrace{\Big\{ (q_i, \tilde{r}_i^{(\le k-1,\mathrm{student})}) \,:\, S_i^{(k-1)} = 1 \Big\}}_{\text{student already correct}} \cup \underbrace{\Big\{ (q_i, r_i^{(k,\mathrm{teacher})}) \,:\, S_i^{(k-1)} = 0, \; C_i^{(k)} = 1 \Big\}}_{\text{regenerate with teacher and filter by correctness}}.$$

---

**Algorithm 1** Adaptive student-aware Distillation for Reasoning

---

**Require: Teacher** $p_L$; **Student** $p_{\text{sm}}^\theta$; **Train set** $\mathcal{D}=\{(q_i, a_i)\}_{i=1}^n$; **Val set** $\mathcal{V}=\{(q_j, a_j)\}_{j=1}^m$; **Instruction** $I$; **Max iters** $K$.

1: Initialize $\theta^{(0)}$ (e.g., from one-step distillation); set history $\mathcal{H} \leftarrow \varnothing$.
2: **for** $k = 1$ **to** $K$ **do**
3:     **// Step 1: Identify learning gaps**
4:     **for** each $(q, a)$ in $\mathcal{V}$ **do**
5:         $r_s \leftarrow \text{GENERATE}(p_{\text{sm}}^{\theta^{(k-1)}}, q, I)$
6:         $score \leftarrow \text{CORRECT}(r_s, a)$               $\triangleright$ 1 if correct, 0 otherwise
7:         $\mathcal{H} \leftarrow \mathcal{H} \cup \{(q, r_s, score)\}$         $\triangleright$ optionally cap to recent $h$ items
8:     **end for**
9:     **// Step 2: Build curated training set (use-student or regenerate)**
10:     $\mathcal{D}_{\text{LLM}}^{(k)} \leftarrow \varnothing$
11:     **for** each $(q, a)$ in $\mathcal{D}$ **do**
12:         $r_s \leftarrow \text{MOSTRECENTSTUDENT}(q)$     $\triangleright$ cached from $k-1$, else $\text{GENERATE}(\cdot)$ once
13:         **if** $\text{CORRECT}(r_s, a)$ **then**             $\triangleright$ student already correct
14:             $\mathcal{D}_{\text{LLM}}^{(k)} \leftarrow \mathcal{D}_{\text{LLM}}^{(k)} \cup \{(q, r_s)\}$         $\triangleright$ *use-student rule*
15:         **else**
16:             $\Delta \leftarrow \text{BUILDGAPINFO}(q, r_s, \text{PREVTEACHER}(q), \mathcal{H})$
17:             $r_t \leftarrow \text{GENERATE}(p_L, q, I, \Delta)$
18:             **if** $\text{CORRECT}(r_t, a)$ **then**         $\triangleright$ accept only if final answer matches
19:                 $\mathcal{D}_{\text{LLM}}^{(k)} \leftarrow \mathcal{D}_{\text{LLM}}^{(k)} \cup \{(q, r_t)\}$
20:             **end if**
21:         **end if**
22:     **end for**
23:     **// Step 3: Student update**
24:     $\theta^{(k)} \leftarrow \text{TRAINSTUDENT}(\theta^{(k-1)}, \mathcal{D}_{\text{LLM}}^{(k)}, I)$     $\triangleright$ standard teacher-forced LM training on curated rationales
25:     **// Step 4: Convergence check**
26:     $val\_acc \leftarrow \text{EVAL}(p_{\text{sm}}^{\theta^{(k)}}, \mathcal{V}, I)$
27:     **if** $\text{NOIMPROVEMENT}(val\_acc)$ **then**
28:         **break**
29:     **end if**
30: **end for**

---

**3) Fine-tune the Student.** We update the student parameters by minimizing the negative log-likelihood of the teacher's new rationales:

$$\mathcal{L}_{\text{L}}^{(k)}(\theta) = -\frac{1}{|\mathcal{D}_{\text{LLM}}^{(k)}|} \sum_{(q_i, r_i^{(k,\text{teacher})}) \in \mathcal{D}_{\text{LLM}}^{(k)}} \sum_{t=1}^{M_i^{(k)}} \log p_{\text{sm}}^\theta\big(r_{i,t}^{(k,\text{teacher})} \,\big|\, r_{i,<t}^{(k,\text{teacher})}, q_i, I\big), \qquad (2)$$

where $M_i^{(k)}$ is the token length of $r_i^{(k,\text{teacher})}$. The updated parameters are

$$\theta^{(k)} = \arg\min_\theta \mathcal{L}_{\text{L}}^{(k)}(\theta). \qquad (3)$$

**4) Convergence Check.** We repeat Steps 1–3 until no further improvement is observed on the validation set or until we reach a maximum iteration $K$.

Algorithm 1 summarizes the proposed adaptive distillation strategy in pseudo code.

## 4 EXPERIMENTAL DETAILS

### 4.1 DATASET

We use the Numina Math dataset (LI et al., 2024) to fine-tune the student models, as each solution is formatted in a Chain-of-Thought (CoT) style. We filter to math *word problems* and further preprocess the corpus to remove samples whose answers are "proof", "notfound", empty, or whose problem/solution fields are invalid. Because Numina Math does not include a validation set, we partition the training set and randomly sample $|\mathcal{V}| = 20$ examples to create a held-out validation set. Since the validation history is kept in the teacher's prompt and grows linearly with iterations, we choose a small validation set so the prompt remains within the teacher's context window even after $K$ iterations.

For testing, we aggregate standard mathematical reasoning benchmarks: GSM8K (Cobbe et al., 2021), MMLU Pro (Wang et al., 2024), MATH 500 (Lightman et al., 2024), and SVAMP (Patel et al., 2021). The overall distribution is shown in Table 1. Additionally, we evaluate out-of-domain (OOD) generalization on StrategyQA (Geva et al., 2021) (687 test samples[2]) and TheoremQA (Chen et al., 2023) (800 test samples), using 4-shot CoT for StrategyQA and 5-shot CoT for TheoremQA.

Table 1: Distribution of training and test datasets.

| Dataset | Training Set (N = 550,000) | | Test Set (N = 4170) | |
|---|---|---|---|---|
| | # Samples | Dist. (%) | # Samples | Dist. (%) |
| **NuminaMath** (LI et al., 2024) | 550,000 | 100% | - | - |
| **MATH 500** (Lightman et al., 2024) | - | - | 500 | 12% |
| **GSM8K** (Cobbe et al., 2021) | - | - | 1319 | 32% |
| **MMLU Pro** (Wang et al., 2024) | - | - | 1351 | 32% |
| **SVAMP** (Patel et al., 2021) | - | - | 1000 | 24% |

### 4.2 MODELS

We employ `llama-3.2-70B` (Dubey et al., 2024) as the teacher model and consider three student models—`llama-3.3-1B`, `qwen-2.5-1.5B` (Bai et al., 2023), and `SmolLM2-1.7B` (Allal et al., 2025). The student models are chosen for their diversity in size and baseline performance. Across experiments, the iterative procedure saturates by $K=4$, with optimal results observed at $K=3$. For each student, we train 5 epochs at iteration $K=1$ and 3 epochs at subsequent iterations. Training each student requires ∼10 GPU hours per epoch on H100 GPUs. Student inference over the training set consumes ∼240 GPU hours on GH200 GPUs, and teacher data generation takes ∼800 GPU hours per iteration on GH200s. Decoding uses vLLM (Kwon et al., 2023) with temperature 0.7.

**Metrics.** We report `acc@1` after numeric answer normalization; "Average" denotes the *micro-average*, i.e., accuracy weighted by per-dataset test counts in Table 1.

## 5 RESULTS AND DISCUSSION

**Iterative adaptive distillation improves student performance.** Table 2 presents the results of three different student models: Qwen2.5 1.5B, SmolLM2 1.7B, and Llama-3.2 1B over three iterations (K) of our iterative distillation procedure AdaptDistill. We report performance on four

---

[2]Test samples taken from https://huggingface.co/datasets/ChilleD/StrategyQA

Table 2: `acc@1` for three student models over iterations (Iter.). Gains (↑/↓) are relative to Iter 1 of the same model (standard distillation). Underlining indicates statistical significant results with $p<0.05$ vs. baseline using a paired test on item-level predictions. "Average" is the micro-average weighted by per-dataset test counts in Table 1.

| Model | Iter. | GSM8K | MATH | MMLU PRO | SVAMP | Average |
|---|---|---|---|---|---|---|
| Qwen | 1 | 50.95 | 32.80 | 14.51 | 76.70 | 43.14 |
| | 2 | 53.22 (↑+2.27) | 35.80 (↑+3.00) | 15.40 (↑+0.89) | 84.20 (↑+7.50) | 46.31 (↑+3.17) |
| | 3 | 55.04 (↑+4.09) | 39.00 (↑+6.20) | 15.17 (↑+0.66) | 87.40 (↑+10.70) | 47.96 (↑+4.82) |
| SmolLM | 1 | 53.60 | 30.60 | 13.47 | 85.50 | 45.49 |
| | 2 | 55.72 (↑+2.12) | 34.20 (↑+3.60) | 15.03 (↑+1.56) | 87.20 (↑+1.70) | 47.51 (↑+2.02) |
| | 3 | 56.93 (↑+3.33) | 38.00 (↑+7.40) | 16.21 (↑+2.74) | 87.20 (↑+1.70) | 48.70 (↑+3.21) |
| Llama | 1 | 34.72 | 20.60 | 9.74 | 79.50 | 35.59 |
| | 2 | 39.34 (↑+4.62) | 24.80 (↑+4.20) | 12.40 (↑+2.66) | 78.50 (↓-1.00) | 38.27 (↑+2.68) |
| | 3 | 43.96 (↑+9.24) | 27.00 (↑+6.40) | 15.06 (↑+5.32) | 80.50 (↑+1.00) | 40.95 (↑+5.36) |

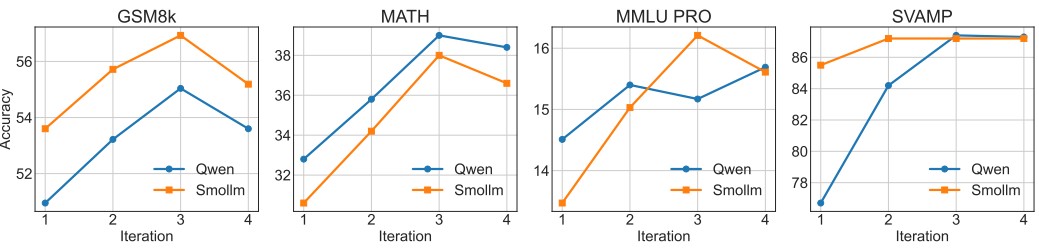

Figure 2: Accuracy (acc@1) for Qwen and SmolLM2 models on GSM8K, MATH, MMLU Pro and SVAMP datasets for 4 iterations.

mathematical reasoning datasets: GSM8K, MATH, MMLU PRO, and SVAMP, along with an Average score across these tasks. We observe a consistent improvement over all datasets across all models. For instance, under Qwen2.5 1.5B, the performance on GSM8K increases from 50.95 at iteration 1 to 55.04 at iteration 3. A similar trend is observed for the MATH dataset, which improves from $32.80 \rightarrow 39.00$, and on SVAMP which increases from $76.70 \rightarrow 87.40$. SmolLM2 and Llama-3.2 also follow the same upward trajectory. Notably, SmolLM2 gains +7.40 points on MATH by iteration 3 (from $30.60 \rightarrow 38.00$) and +2.74 points on MMLU PRO (from $13.47 \rightarrow 16.21$). Llama-3.2 sees gains of +9.24 and +8.40 points on GSM8K and MATH, respectively, moving from 34.72 $\rightarrow$ 43.96 on GSM8K and 20.60 $\rightarrow$ 29.00 on MATH. Overall, each model consistently benefits from iterative distillation, achieving higher accuracy on each dataset from iteration 1 to iteration 3.

**Deciding when to stop.** Figure 2 visualizes Qwen2.5 and SmolLM2 across four iterations on the same four benchmarks, mirroring the upward trend through Iter 3 in Table 2. At Iter 4, accuracy slightly declines (e.g., Qwen2.5: $55.04 \rightarrow 53.60$ on GSM8K; SmolLM2: $56.93 \rightarrow 55.19$ on GSM8K), indicating convergence. Since these iteration 4 numbers are lower than iteration 3 results, we conclude that the model has converged and additional regeneration from the teacher is no longer beneficial, prompting us to stop further distillation steps at iteration 3.

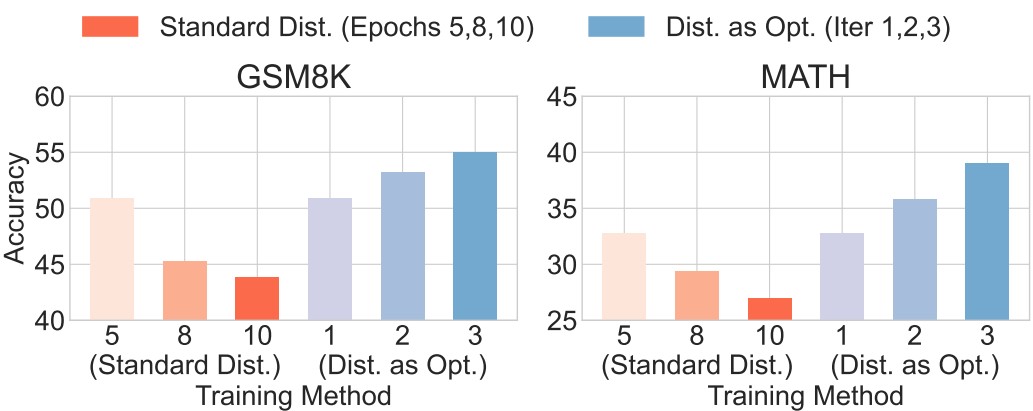

Figure 3: Accuracy comparison of training standard distillation and AdaptDistill for the equal number of epochs. Note that iteration 1 for AdaptDistill is same as epoch 5 for standard distillation.

Table 3: Comparison of `acc@1` accuracy for Qwen and SmolLM across four benchmarks (GSM8K, MATH, MMLU PRO, SVAMP) and the resulting average score. Each model can be trained on data generated by its own teacher or a cross-teacher (e.g., Qwen data for SmolLM2). Gains ($\uparrow$ or $\downarrow$) are shown relative to the baseline (standard distillation). Underline indicates statistical significance with $p < 0.05$ compared to the baseline.

| | Teacher | GSM8K | MATH | MMLU PRO | SVAMP | Average |
|---|---|---|---|---|---|---|
| **Qwen** | Baseline | 50.95 | 32.80 | 14.51 | 76.70 | 43.14 |
| | Qwen | 56.10 (↑+5.15) | 39.40 (↑+6.60) | 15.98 (↑+1.47) | 86.50 (↑+9.80) | 48.39 (↑+5.52) |
| | SmolLM2 | 54.00 (↑+3.05) | 34.00 (↑+1.20) | 14.35 (↓-0.16) | 86.50 (↑+9.80) | 46.57 (↑+3.43) |
| **SmolLM** | Baseline | 53.60 | 30.60 | 13.47 | 85.50 | 45.49 |
| | Qwen | 56.86 (↑+3.26) | 32.80 (↑+2.20) | 14.95 (↑+1.48) | 85.60 (↑+0.10) | 47.29 (↑+1.80) |
| | SmolLM2 | 56.86 (↑+3.26) | 33.60 (↑+3.00) | 15.00 (↑+1.53) | 86.80 (↑+1.30) | 47.72 (↑+2.23) |

**Can standard distillation trained for same number of epochs perform similar to AdaptDistill?** Figure 3 compares standard distillation (continued epochs) to AdaptDistill under the same total number of epochs. Extending standard training (e.g., 10 epochs) yields diminished returns and overfitting (GSM8K 43.90%, MATH 27.00%, MMLU Pro 13.82%, SVAMP 72.90%, average 39.41%), all below the Iter 1 baseline. In contrast, AdaptDistill improves each round: for Qwen2.5, GSM8K 50.95% → 55.04%, MATH 32.80% → 39.00%, MMLU Pro 14.51% → 15.17%, SVAMP 76.70% → 87.40%, raising the average from 43.14% to 47.96%. Iterative, gap-aware regeneration is therefore more effective than simply training longer.

**Does AdaptDistill create a better *teacher-generated dataset*?** Table 3 examines training each student for 5 epochs on datasets generated at the *final* iteration of the teacher. The column "Teacher" denotes which student's gap signals were used to condition the (fixed) `llama-3.2-70B` teacher during data generation ("Baseline" is standard distillation). For Qwen2.5, using Qwen-conditioned teacher generations yields sizable gains (GSM8K 50.95 → 56.10, SVAMP 76.70 → 86.50, average +5.52). Even SmolLM2-conditioned teacher data improves Qwen over baseline (average +3.43). SmolLM2 exhibits the same pattern: with SmolLM2-conditioned teacher data, GSM8K rises to 56.86 and MMLU Pro to 15.00, while Qwen-conditioned data also helps (average +1.80). These

Table 4: Performance (%) on **StrategyQA** and **Theorem QA** benchmarks for Qwen and SmolLM2 models. Standard Distillation represents the results from Iteration 1 while AdaptDistill represents the results from Iteration 3 from Table 2. Gains (↑) are shown in parentheses.

| Model | StrategyQA | | Theorem QA | |
|---|---|---|---|---|
| | **Standard Dist.** | **AdaptDistill** | **Standard Dist.** | **AdaptDistill** |
| **Qwen** | 57.50% | 60.00% (↑+2.50) | 7.5% | 8.125% (↑+0.625) |
| **SmolLM2** | 60.20% | 63.00% (↑+2.80) | 5.62% | 6.25% (↑+0.63) |

results suggest the iterative procedure produces teacher outputs that are *personalized* to a student's weaknesses and transfer partly across students.

**Does AdaptDistill cause overfitting to mathematical task?** To assess overfitting, we evaluate on out of domain (OOD) datasets: StrategyQA and TheoremQA. Table 4 shows Qwen and SmolLM2 both improve from Iter 1 (standard distillation) to Iter 3 (AdaptDistill) on **both** OOD tasks. For Qwen: +2.50% on StrategyQA and +0.625% on TheoremQA; for SmolLM2: +2.80% and +0.63%, respectively. These gains suggest that AdaptDistill enhances general reasoning rather than overfitting to mathematical problems. We hypothesize that mathematical reasoning serves as a structured training signal. The explicit CoT traces learned in math tasks generalize to tasks requiring multi-hop reasoning and factual consistency, as observed on StrategyQA. Our aim here is not to maximize absolute OOD scores, but to test whether the adaptive procedure compromises out-of-domain performance; empirically, it does not.

# 6 CONCLUSION

We introduced an iterative distillation paradigm AdaptDistill aimed at bridging the performance gap between large teacher models and smaller student models more effectively than conventional one-shot distillation. By repeatedly identifying the student's errors, regenerating teacher rationales that address these *learning gaps*, and re-training the student on this tailored data, our method yields consistent improvements on challenging mathematical benchmarks. Experimental results indicate that AdaptDistill not only avoids overfitting often seen in prolonged standard distillation, but also makes the teacher more attuned to student-specific weaknesses, effectively "personalizing" the teacher's rationales. Empirically, we observed that final teacher-generated data after multiple rounds of iteration significantly enhances student performance, with gains of up to 20% compared to the baseline. Even cross-teacher data (e.g., teacher data for one student used on another) shows improved results over standard approaches. These findings underscore that incorporating iterative feedback loops between the teacher and the student leads to a more specialized and effective transfer of reasoning knowledge.

# 7 ETHICS STATEMENT

This work uses math problems from publicly available datasets such as NuminaMath, GSM8K, MATH, SVAMP and MMLU Pro for fine-tuning and evaluation. These datasets are widely used in prior research and do not involve sensitive or personally identifiable information. The validation set for our prompts was also created from the same sources.

We acknowledge that large language models may produce harmful or unsafe outputs; developing safeguards to mitigate such risks remains important future work. All references have been properly cited, and we have adhered to standard research integrity and ethical practices throughout this project.

## 8 REPRODUCIBILITY STATEMENT

We have taken several measures to ensure reproducibility:

- **Datasets:** We use only publicly available datasets (NuminaMath, GSM8K, MATH, MMLU Pro, SVAMP), with preprocessing and prompts described in section 4.1 and appendix A.
- **Models:** The teacher and student model architectures are explicitly stated (LLaMA-3.2-70B: **teacher**; Qwen-2.5-1.5B, SmolLM2-1.7B, LLaMA-3.3-1B: **students**).
- **Training Setup:** Hyperparameters, iteration schedules, and optimization details are fully specified in section 4.2.
- **Compute Resources:** We report GPU hours and hardware details used for both student training and teacher generation in section 4.2.
- **Code Availability:** All source code, data preprocessing scripts, and evaluation pipelines will be released upon publication to facilitate exact reproduction.

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

## A  APPENDIX

**Prompt file given to the teacher in the first iteration.**

---

```
You are an AI tutor tasked with improving a student's understanding of
mathematical problem-solving. You will be given a question, a teacher's
answer, a student's answer, and a score. Your job is to analyze these
inputs and create a new answer that will help the student learn better.

Here are some examples of the task:

### question:
Jim spends 8 hours scuba diving. In that time he finds a treasure chest
with 100 gold coins in it. He also finds some smaller bags that have half
as much gold each. He finds 25 gold coins per hour. How many smaller bags
did he find?
### teacher answer:
Jim finds 25 gold coins per hour, and he spends 8 hours scuba diving, so
he finds a total of 25 * 8 = 200 gold coins.
He finds a treasure chest with 100 gold coins in it, so the remaining
gold coins he finds are 200 - 100 = 100 gold coins.
Since the smaller bags have half as much gold as the treasure chest, each
smaller bag has 100 / 2 = 50 gold coins.
Therefore, the number of smaller bags he finds is 100 / 50 = $\boxed{2}$
smaller bags.
Final Answer: 2
### student answer:
To determine how many smaller bags Jim found, we need to follow these
steps:
1. **Calculate the total amount of gold coins Jim finds over the 8
hours:** Jim finds 25 gold coins per hour for 8 hours. \[ 25 \text{
coins/hour} \times 8 \text{ hours} = 200 \text{ coins} \]
2. **Determine the amount of gold coins in the smaller bags:** Let \( x
\) be the number of smaller bags. Each smaller bag has half as much gold
as a treasure chest. Since a treasure chest has 100 gold coins, each
smaller bag has: \[ \frac{100 \text{ coins}}{2} = 50 \text{ coins} \]
Therefore, the total amount of gold coins in the smaller bags is: \[ 50x
\text{ coins} \]
3. **Set up the equation for the total gold coins found:** The total gold
coins found by Jim is the sum of the gold coins in the treasure chest and
the gold coins in the smaller bags. According to the problem, this total
is 200 coins. \[ 100 \text{ coins} + 50x \text{ coins} = 200 \text{
coins} \]
```

4. **Solve for \( x \):** Subtract 100 coins from both sides of the equation: \[ 50x = 100 \] Divide both sides by 50: \[ x = \frac{100}{50} = 2 \] Thus, Jim found \(\boxed{2}\) smaller bags.
Final Answer: 2
### score:
1

------------similar such examples from the validation set------------

First, carefully analyze the student's answer. Compare it to the teacher's answer and identify any mistakes or areas where the student's reasoning could be improved. Consider the following:

1. Did the student understand the problem correctly?
2. Did they use the right approach to solve the problem?
3. Are there any calculation errors?
4. Is their reasoning clear and logical?
5. Did they miss any important steps?

Next, craft a new answer that addresses the student's misunderstandings or reinforces correct thinking. Your new answer should:

1. Use clear, step-by-step reasoning
2. Explain any concepts the student may have misunderstood
3. Provide additional context or examples if necessary
6. Use the same calculation format as the teacher's answer. If the teacher answer involves latex and involves terms like \frac, \pi make sure to extract the answer with the necessary latex keywords.
5. All the teacher's final answers end inside "\boxed{}". Ensure, that your answers also follow this format.
6. Lead to the correct final answer

Write your new answer using the following format:

### new_answer
[Step-by-step reasoning with calculations in the format shown above]
Final Answer: [Correct numerical/latex answer]

Remember, your goal is to help the student learn and improve their problem-solving skills. Focus on explaining the reasoning clearly and addressing any specific issues in the student's original answer.

---

**Prompt file given to the teacher in the second iteration.**

---

You are an AI tutor tasked with improving a student's understanding of mathematical problem-solving. You will be given a question, a teacher's answer, a student's answer, and a score. Your job is to analyze these inputs and create a new answer that will help the student learn better.

Here are some examples of the task:

### question:
Jim spends 8 hours scuba diving. In that time he finds a treasure chest with 100 gold coins in it. He also finds some smaller bags that have half as much gold each. He finds 25 gold coins per hour. How many smaller bags did he find?
### ITERATION 1:
### teacher answer:
Jim finds 25 gold coins per hour, and he spends 8 hours scuba diving, so he finds a total of 25 * 8 = 200 gold coins.
He finds a treasure chest with 100 gold coins in it, so the remaining gold coins he finds are 200 - 100 = 100 gold coins.
Since the smaller bags have half as much gold as the treasure chest, each smaller bag has 100 / 2 = 50 gold coins.
Therefore, the number of smaller bags he finds is 100 / 50 = $\boxed{2}$ smaller bags.
Final Answer: 2
### student answer:
To determine how many smaller bags Jim found, we need to follow these steps:
1. **Calculate the total amount of gold coins Jim finds over the 8 hours:** Jim finds 25 gold coins per hour for 8 hours. \[ 25 \text{ coins/hour} \times 8 \text{ hours} = 200 \text{ coins} \]
2. **Determine the amount of gold coins in the smaller bags:** Let \( x \) be the number of smaller bags. Each smaller bag has half as much gold as a treasure chest. Since a treasure chest has 100 gold coins, each smaller bag has: \[ \frac{100 \text{ coins}}{2} = 50 \text{ coins} \]
Therefore, the total amount of gold coins in the smaller bags is: \[ 50x \text{ coins} \]
3. **Set up the equation for the total gold coins found:** The total gold coins found by Jim is the sum of the gold coins in the treasure chest and the gold coins in the smaller bags. According to the problem, this total is 200 coins. \[ 100 \text{ coins} + 50x \text{ coins} = 200 \text{ coins} \]
4. **Solve for \( x \):** Subtract 100 coins from both sides of the equation: \[ 50x = 100 \] Divide both sides by 50: \[ x = \frac{100}{50} = 2 \] Thus, Jim found \(\boxed{2}\) smaller bags.
Final Answer: 2
### score:
1
### ITERATION 2:
### teacher answer:
To determine how many smaller bags Jim found, we need to follow these steps:
1. **Calculate the total amount of gold coins Jim finds over 8 hours:** Jim finds 25 gold coins per hour for 8 hours. The total gold coins he finds is: \[ 25 \times 8 = 200 \text{ gold coins} \]
2. **Determine the amount of gold coins in the smaller bags:** The problem states that each smaller bag has half as much gold as a treasure chest. Since a treasure chest contains 100 gold coins, a smaller bag will have: \[ \frac{100}{2} = 50 \text{ gold coins} \]

3. **Set up an equation to find the number of smaller bags \( x \):** The total amount of gold coins found in the smaller bags over 8 hours should equal the total gold coins found minus the gold coins in the treasure chest. Let \( x \) be the number of smaller bags found. The total gold coins found in the smaller bags is: \[ 50x \] This total must equal the remaining gold coins after subtracting the treasure chest: \[ 50x = 200 - 100 \] \[ 50x = 100 \]

4. **Solve for \( x \):** To find \( x \), divide both sides of the equation by 50: \[ x = \frac{100}{50} = 2 \] Thus, the correct answer is: \boxed{2} Final Answer: \boxed{2}

Final Answer: 2

### student answer:

To find the number of smaller bags Jim found, we first calculate the total amount of gold coins he has.

Since he finds 25 gold coins per hour for 8 hours, the total amount of gold coins he finds is $25 \times 8 = 200$ gold coins. Next, we determine the amount of gold coins in each smaller bag. Each smaller bag has half as much gold as a treasure chest, which is $\frac{100}{2} = 50$ gold coins.

Now, we divide the total amount of gold coins found by the amount in each smaller bag to find the number of smaller bags: $\frac{200}{50} = 4$. Therefore, Jim found $\boxed{4}$ smaller bags. Final Answer: \boxed{4}.

Final Answer: 4

### score:

0

------------similar such examples from the validation set------------

First, carefully analyze the student's answer. Compare it to the teacher's answer and identify any mistakes or areas where the student's reasoning could be improved. Consider the following:

1. Did the student understand the problem correctly?
2. Did they use the right approach to solve the problem?
3. Are there any calculation errors?
4. Is their reasoning clear and logical?
5. Did they miss any important steps?

Next, craft a new answer that addresses the student's misunderstandings or reinforces correct thinking. Your new answer should:

1. Use clear, step-by-step reasoning
2. Explain any concepts the student may have misunderstood
3. Provide additional context or examples if necessary
6. Use the same calculation format as the teacher's answer. If the teacher answer involves latex and involves terms like \frac, \pi make sure to extract the answer with the necessary latex keywords.
5. All the teacher's final answers end inside "\boxed{}". Ensure, that your answers also follow this format.
6. Lead to the correct final answer

Write your new answer using the following format:

### new_answer

```
[Step-by-step reasoning with calculations in the format shown above]
Final Answer: [Correct numerical/latex answer]
```

Remember, your goal is to help the student learn and improve their
problem-solving skills. Focus on explaining the reasoning clearly and
addressing any specific issues in the student's original answer.

