# OpenReview forum: "Teachers That Listen: Adaptive Student-Aware Distillation for Reasoning"
_ICLR.cc/2026/Conference — Submitted to ICLR 2026_

### Official Review · Reviewer_A1UH · 2025-11-01

**Soundness:** 3
**Presentation:** 2
**Contribution:** 2
**Rating:** 4
**Confidence:** 4

**Summary:**

The core idea of this work is to reframe knowledge distillation as an iterative teacher-student interaction process, aiming to bridge the gap between teacher-generated generic reasoning chains (Rationales) and the specific learning needs of the student model. In each iteration, the teacher first generates initial rationales. The student model learns from them, attempts the task, and reveals its learning difficulties and errors. Subsequently, the teacher refines and regenerates explanations specifically, based on the student model's error feedback and historical performance. The student model is ultimately fine-tuned on a curated dataset mixed with its own correct reasoning traces and the teacher's refined traces.Empirical results show that the AdaptDistill approach achieves significant performance gains compared to standard one-shot distillation methods on mathematical and common-sense reasoning tasks, including GSM8K and MATH.

**Strengths:**

1) The paper proposes a novel closed-loop adaptive distillation framework. By explicitly reintroducing the student model's error feedback into the teacher's generation step, it achieves customized guidance targeting individual student weaknesses, effectively addressing the mismatch between teacher output and student needs in standard distillation.
2) The method achieves a performance increase of up to 20% in accuracy on several challenging reasoning tasks compared to basic one-shot distillation baselines.
3) Experiments demonstrate that the method not only improves accuracy on tasks within the training distribution but also maintains and improves performance on Out-of-Domain (OOD) tasks (such as StrategyQA and TheoremQA), indicating an enhancement in the transferability of reasoning skills.

**Weaknesses:**

1) The core technical component of this method relies on the powerful teacher model (Llama-3.2-70B) performing the steps of "identifying learning gaps" and "generating customized, refined explanations" via Prompt Engineering. Although the framework concept is novel, the mechanism for "identifying learning gaps" and "refining explanations" lacks a quantifiable, learnable, modular design. Instead, it relies on the black-box reasoning capability of a Large Language Model (LLM) to act as an "in-context optimizer." This diminishes the purely technical innovation of the method itself and poses difficulties for future research in reproduction and improvement.
2) The paper's primary comparative baseline is "Standard One-Shot Knowledge Distillation" (i.e., performing CoT distillation in a single pass, following Shridhar et al. (2023)). This is overly simplistic given the current advancements in the knowledge distillation field. As you pointed out, the paper fails to compare against several recent and comparable methods. It lacks empirical comparison against recent State-of-the-Art (SOTA) works like DistilLLM and MiniLLM, which makes it difficult to conclusively prove AdaptDistill's leading position at the current technological frontier. While the paper mentions several iterative or adaptive distillation methods in the related work section (e.g., Wang et al. (2023), Adarsh et al. (2025), Agarwal et al. (2024)), it does not conduct direct performance comparisons between AdaptDistill and these iterative methods that are most similar in mechanism, making it difficult to fully demonstrate the superiority of its unique teacher feedback mechanism.
3) The validation set V used in the paper consists of only 20 samples. The authors justify this by the need to ensure the teacher model's context window can accommodate the historical records H for all iterations. However, using the historical performance of a set of only 20 samples to guide the teacher in targeted content generation over the entire training set $\mathcal{D}$ for multiple rounds may lead to biased guidance or cause the teacher's refinement process to overfit to these 20 samples, thereby affecting the final student model's generalization and robustness.

**Questions:**

1) Given that your method is iterative and adaptive reasoning distillation, please supplement the experimental results with performance comparisons against representative similar works, such as Wang et al. (2023) or Adarsh et al. (2025), which you cited in your related work. Please also explain why comparisons with DistilLLM Ko et al. (2024) and MiniLLM Gu et al. (2023), which are established or recent works in knowledge distillation, were omitted. If there are technical differences that make direct comparison infeasible, please clarify this in the paper.
2) The "identifying learning gaps" and "generating refined explanations" steps are crucial to AdaptDistill, primarily implemented via prompting the teacher model. Please publicly release the complete Prompt templates used to generate the gap information and the refined rationale in the Appendix. This is necessary to ensure the reproducibility of the experiments and allow readers to better understand the teacher model's decision mechanism.
3) Please provide a more in-depth discussion on the choice of using only 20 samples for the validation set. Furthermore, conduct an ablation study to compare the performance when using a larger validation set (e.g., 100 or 200 samples) under the constraint that the historical record H is limited to a small sliding window (e.g., only considering the 20 most recent samples instead of accumulating all history), to verify the robustness and representativeness of the current small validation set.

---

> ### Author Response · Authors · 2025-11-18
> **Rebuttal**
>
> We thank the reviewer for their thoughtful assessment and for clearly recognizing the key strengths of our work: (i) proposing a **novel closed-loop adaptive distillation framework**, (ii) achieving up to **20% accuracy gains** over basic one-shot distillation on challenging reasoning tasks, and (iii) improving **out-of-domain (OOD) performance**, indicating enhanced transferability of reasoning skills.
>
> Below we address each concern in turn and respond to the specific questions.
>
> ## **Concern 1: On using the teacher as an “in-context optimizer”**
>
> Since identifying “learning gaps” and “refined explanations” stages are central to AdaptDistill and at the same time we believe that **relying on a strong LLM teacher here is both unavoidable and appropriate** as the nature of the tasks are in the **natural language reasoning setting**. The task is not simply “label correction” but **diagnosing intermediate reasoning failures** in arbitrary natural language traces produced by the student and the student can make mistakes in many ways (skipping steps, misapplying formulas, misinterpreting text). Codifying all such patterns in a **rule-based or purely symbolic module is intractable at scale**. Using a large language model as a proxy for a human teacher is important to understanding the student’s reasoning trace, comparing it with a correct solution and generating a **pedagogically adapted rationale** that is precisely what makes the closed-loop process feasible.
>
> Our **technical novelty is not “we prompt a teacher”, but the full framework around that**, including:
>
> 1. A **closed-loop, history-aware interaction between teacher and student**, where the teacher is explicitly conditioned on the trajectory of student errors and scores, not just a single failed attempt and the teacher builds an implicit teaching strategy from this history, rather than issuing isolated corrections.
> 2. A **curated training mix** of student own correct reasoning trace and teacher’s history-adapted rationales targeted at persistent student weaknesses.
> 3. A clear, reproducible protocol that can be **plugged into other teacher models or future learned modules** (e.g., a trainable “gap-identification” head) without changing the core loop.
>
> To directly address the reproducibility and future extensibility, **we already release the full prompt templates**. So while we do rely on a powerful teacher LLM, this is fundamentally the same assumption made by all modern distillation methods; what AdaptDistill adds is a general and reusable closed-loop framework for turning that teacher into an adaptive instructor rather than a one-shot label generator.
>
> ## **Concern 2: comparison with other baselines**
>
> We fully agree that a serious distillation paper must go beyond basic one-shot CoT as a baseline. Based on your concern, we have **run additional experiments to compare AdaptDistill against several strong and closely related methods**, particularly iterative and student-/error-aware ones.
>
> ### We compare AdaptDistill against:
> - **Direct CoT baselines** (e.g., Socratic CoT style).
> - **Iterative, single-teacher distillation** (SIKeD, Adarsh et al.).
> - **Non-iterative, single-/multi-teacher methods** (Learning from Committee, Li et al.).
> - **Non-iterative, multi-strategy methods** (Mixed Distillation, Li et al.).
>
>
> ### Experimental setup.
> - **Dataset**: 1000 SVAMP math samples with modified questions that are particularly challenging for small models.
> - **Student**: Qwen2.5-1.5B-Instruct used for all approaches for a strictly controlled comparison.
> - **Teacher**: As indicated in the table below (single GPT, committees, single llama, etc.).
>
> ## Results (SVAMP, 1000 samples)
>
> | Approach                           | Teacher                          | Iterative? | Accuracy          |
> |------------------------------------|----------------------------------|-----------:|-------------------|
> | Learning from Committee – Li et al.| Single (GPT)                     | No         | 79.33             |
> | Learning from Committee – Li et al.| Multiple (GPT, Mixtral, Gemini)  | No         | 77.67             |
> | Learning from Committee – Li et al.| Multiple + peer review           | No         | 81.00             |
> | Mixed Distillation – Li et al.     | Single                           | No         | 73.20             |
> | SIKeD – Adarsh et al.              | Single (Llama)                   | Yes        | 75.40             |
> | **AdaptDistill (ours)**                | Single (Llama)                   | Yes        | **87.40 (+7 to +15 pts)** |
>
>
> **AdaptDistill substantially outperforms all stronger baselines** , including recent multi-teacher methods and sophisticated iterative or mixed strategies **by +7–15 absolute accuracy** points on a challenging reasoning benchmark.
>
> This improvement is obtained **under the same student model** and comparable teacher setups, meaning the gain is attributable to the distillation procedure itself, not to a stronger student or teacher.

---

> > ### Author Response · Authors · 2025-11-18
> > **Rebuttal Continued (II)**
> >
> > ## **Concern 3: limited validation set and potential bias**
> >
> > We designed the validation set and teacher prompts specifically to mitigate biases if any from the framework.
> >
> > 1. We created a careful diverse set of 20 validation examples which are not random. They are manually sampled to cover:
> >   - **Different difficulty levels** (from simpler SVAMP-style problems to complex MATH/MMLU items),
> >   - **Different reasoning lengths** (short vs. long CoT),
> >   - **Different domains** (single step vs multi-step reasoning).
> >
> > This means the teacher is not seeing 20 near-identical problems, but a **diverse probe set** representing various failure modes.
> >
> > 2. We must *keep all iterations’ history* for these 20 samples in the teacher’s context. Complex tasks like MATH and MMLU have extremely long reasoning chains, so a significantly **larger validation set would not fit in the teacher’s context** across iterations. We cannot truncate it as truncating history or dropping long examples would weaken the very signal (error trajectories) that AdaptDistill is built on.
> >
> > 3. **Teacher uses the validation set as a probe, not as training data**. The student **never trains** on the validation set; it is only used to measure how student performance evolves over iterations, and to provide **high-level understanding to the teacher** about which strategies work or fail. In other words, the teacher uses these 20 items as a **diagnostic panel**, not as the full representation of the training data. This greatly reduces the risk of the student overfitting to them.
> >
> > 4. If the validation set were causing severe bias or overfitting, **we would expect poor generalization on held-out test sets,** or a degradation on OOD tasks like StrategyQA and TheoremQA. **Instead we observe consistent gains** on in-domain test sets, and improved performance on OOD benchmarks, which strongly suggests that **the teacher’s guidance is not overfitting** to these 20 samples but improving general reasoning.
> >
> > # Questions
> >
> > ## **Compare with past work**
> > We have precisely compared our work with past work mentioned in details above. Please refer to the table above. In summary, all under t**he same student and comparable teacher configurations*3, showing **+7–15 point gains for AdaptDistill**.
> >
> > ## **Prompts in the Appendix**
> > We have **already provided the prompts in the Appendix**. Please have a look and let us know if for some reason it is missing for you. We have provided **prompts for both the first iteration and for the subsequent iterations**.
> >
> > ## **Validation set**
> >
> > We have discussed this in detail above about how the examples were selected, how their diversity is important to prevent any biases and how context window limitations do not allow us to go beyond 20 samples.
> >
> > Finally we want to clarify that we don't keep the last 20 samples in the context window but it is a **held out validation set** that we always use. This is important for the model to track the history of its own generation and how the student performance has changed over time with different strategies deployed.
> >
> > Given these clarifications and new results, we believe AdaptDistill represents a solid and practically important advance in reasoning distillation rather than a marginal contribution. We respectfully ask the reviewer to reconsider their overall rating and raise the score accordingly.
> >
> > Please let us know if we answered your concerns. We are happy to discuss more.

---

> > > ### Author Response · Authors · 2025-11-24
> > > **Acknowledgment of the rebuttal**
> > >
> > > We thank all reviewers for their time and insightful comments.
> > > In the rebuttal above, **we have addressed the main concerns regarding **the validation set used, comparison of our work with other baselines** and clarified several other points that were previously unclear.
> > >
> > > If our clarifications and additional experiments resolved your concerns, we would be very grateful if you could acknowledge this and also if you could consider updating your review to reflect your current assessment.
> > >
> > > Thank you so much.
> > >
> > > Please let us know if we can clarify anything else.

---

### Official Review · Reviewer_w69m · 2025-11-01

**Soundness:** 3
**Presentation:** 2
**Contribution:** 3
**Rating:** 4
**Confidence:** 2

**Summary:**

The Adaptive student-aware Distillation for Reasoning (AdaptDistill) is designed to bridge this gap by iteratively identifying the student’s errors and allowing the teacher to refine its explanations according to the student’s needs.

**Strengths:**

1. The paper proposes an innovative solution to the distributional mismatch between the teacher’s rationales and the student’s learning bottlenecks, effectively enhancing instructional alignment.

2. The framework is rigorously defined and the experiments are carefully designed, ensuring the reliability and reproducibility of the results.

3. The proposed method shows strong potential to significantly improve the distillation process, especially for tasks involving complex reasoning.

**Weaknesses:**

- Lack of Ablation Studies: While the method shows strong results, the paper could benefit from more ablation studies comparing AdaptDistill with other state-of-the-art iterative distillation methods. This would provide a clearer picture of how AdaptDistill fares relative to similar approaches.
- Limited Task Variety: The experiments primarily focus on mathematical and commonsense reasoning tasks. While these are important, the paper could be strengthened by demonstrating the method’s effectiveness across a broader range of tasks or domains.
- Scalability Concerns: The iterative nature of the method requires multiple rounds of distillation, which can become computationally expensive. The paper could discuss strategies for making this process more efficient or scalable, particularly when applying it to larger datasets or models.
- Model Transferability: Although the paper tests AdaptDistill on different student models, further exploration into the transferability of the learned knowledge across different architectures would be valuable. It would also be useful to understand how AdaptDistill performs with varying model sizes.

**Questions:**

1. Could the authors provide more details on the scalability of AdaptDistill? How would it perform with much larger datasets or models?
2. How does AdaptDistill compare to other advanced distillation methods, such as reinforcement learning-based or self-guided distillation approaches?
3. What are the potential limitations of the iterative refinement process in terms of model convergence and overfitting after many iterations?
4. How would AdaptDistill perform on tasks beyond the domains tested, particularly on tasks that require high levels of generalization?

---

> ### Author Response · Authors · 2025-11-18
> **Rebuttal**
>
> We thank the reviewer for **their positive assessment of our work**, specifically, for recognizing that **AdaptDistill offers an innovative solution** to the distributional mismatch between teacher rationales and student bottlenecks, that **the framework is rigorously defined with carefully designed experiments,** and that it has **strong potential for improving distillation** on complex reasoning tasks.
>
>
> Below, we address the concerns on ablations, task variety, scalability, and transferability, and respond to the reviewer’s questions.
>
> ## **Concern 1: Comparison to other distillation approaches**
>
> We agree this is an important point and have run additional experiments specifically to address it. In particular, **we compare AdaptDistill against several strong and conceptually related distillation baselines**, including iterative and student-/error-aware methods.
>
>
> ### We compare AdaptDistill against:
> - **Direct CoT baselines** (e.g., Socratic CoT style).
> - **Iterative, single-teacher distillation** (SIKeD, Adarsh et al.).
> - **Non-iterative, single-/multi-teacher methods** (Learning from Committee, Li et al.).
> - **Non-iterative, multi-strategy methods** (Mixed Distillation, Li et al.).
>
>
> ### Experimental setup.
> - **Dataset**: 1000 SVAMP math samples with modified questions that are particularly challenging for small models.
> - **Student**: Qwen2.5-1.5B-Instruct used for all approaches for a strictly controlled comparison.
> - **Teacher**: As indicated in the table below (single GPT, committees, single llama, etc.).
>
> ## Results (SVAMP, 1000 samples)
>
> | Approach                           | Teacher                          | Iterative? | Accuracy          |
> |------------------------------------|----------------------------------|-----------:|-------------------|
> | Learning from Committee – Li et al.| Single (GPT)                     | No         | 79.33             |
> | Learning from Committee – Li et al.| Multiple (GPT, Mixtral, Gemini)  | No         | 77.67             |
> | Learning from Committee – Li et al.| Multiple + peer review           | No         | 81.00             |
> | Mixed Distillation – Li et al.     | Single                           | No         | 73.20             |
> | SIKeD – Adarsh et al.              | Single (Llama)                   | Yes        | 75.40             |
> | **AdaptDistill (ours)**                | Single (Llama)                   | Yes        | **87.40 (+7 to +15 pts)** |
>
>
> **AdaptDistill substantially outperforms all stronger baselines** , including recent multi-teacher methods and sophisticated iterative or mixed strategies **by +7–15 absolute accuracy** points on a challenging reasoning benchmark.
>
> This improvement is obtained **under the same student model (Qwen2.5-1.5B-Instruct)** and comparable teacher setups, meaning the gain is attributable to the distillation procedure itself, not to a stronger student or teacher.
>
> ## **Concern 2: Task variety**
>
> While we agree that broad task coverage is important, our experiments **already go significantly beyond “just math”** and are in line with prior distillation work. We evaluate on six datasets spanning different difficulty regimes and domains:
> - **SVAMP, GSM8K** – grade-school style math and word problems.
> - **MATH, MMLU** – considerably more challenging, high-level mathematical and multi-domain reasoning.
> - **StrategyQA, TheoremQA** – commonsense and knowledge-intensive reasoning.
>
> This suite was chosen deliberately to **align with and extend past distillation work** (e.g., Magister et al., Shridhar et al., Adarsh et al., Li et al.) while covering simple to difficult reasoning (other works are missing challenging datasets like MATH and MMLU), numerical vs common sense reasoning, and in domain vs out of domain datasets (also missing from a lot of past works).
> Moreover, in our OOD experiments, **AdaptDistill improves performance even on tasks it is not explicitly trained on**, indicating that it **enhances general reasoning ability, not just overfits to particular math benchmarks**.
>
> Given page limits and established practice in the area, we believe this breadth is sufficient to demonstrate that AdaptDistill is not limited to one narrow setting, and we will make the cross-domain nature of our evaluation more explicit in the revision.

---

> > ### Author Response · Authors · 2025-11-18
> > **Rebuttal continued (II)**
> >
> > ## **Concern 3: Scalability and computational cost**
> >
> > We fully agree that iterative distillation is more computationally demanding than a one-shot pass. This is a fundamental property shared with all student-on-policy and multi-iteration distillation methods. Our goal is not to propose the cheapest possible distillation, but rather a **substantially more effective one for a given teacher–student pair**.
> >  That said, we do take cost into account:
> > - We **never re-query the teacher for items the student already gets correct**. Those examples are simply reused as student-generated training data, reducing unnecessary teacher calls.
> > - To stay fair, we match the training compute for baselines (e.g., standard distillation trained for the same number of epochs). Despite this, standard baselines perform way worse than AdaptDistill.
> >
> > **In terms of scaling to larger datasets and models**:
> > - The procedure of AdaptDistill is very **parallel across training instances** : student forward + correctness check + teacher regeneration for failures can be distributed over shards of data.
> > - In practice, the number of iterations can be **capped** (e.g., 2–3 iterations) once performance saturates, which we already observe in our experiments.
> >
> > ## **Concern 4: Model transferability and cross-architecture behavior**
> >
> > We agree that understanding transfer across model architectures and sizes is valuable. While **exhaustively sweeping many architectures is beyond the current scope** (and often constrained by proprietary training stacks), we already provide evidence that **AdaptDistill is not tied to a single model pairing** by experimenting with different teacher–student combinations, not just a single fixed pair.
> >
> > In particular, able 3 in the paper shows that each student trained on data generated by its own teacher, and also in a **cross-teacher** setting, where a different teacher generates the distillation data. We find that **cross-teacher AdaptDistill still outperforms baselines**, indicating that the iterative procedure produces teacher outputs that are personalized to a student’s weaknesses but remain partly **transferable across students**.
> >
> > Finally, we have conducted experiments with student models in the 1B–2B parameter range, and see **consistent improvements when scaling to the larger student within this range**. This matches the common distillation practice where models like LLaMA-1B or Gemma-2B are distiled from larger, stronger teachers. AdaptDistill slots naturally into that standard pipeline.

---

> > > ### Author Response · Authors · 2025-11-18
> > > **Rebuttal Continued (III)**
> > >
> > > # Questions
> > >
> > > ## **Scalability of the approach**
> > >
> > > As discussed in the last point above, the method is quite scalable.
> > > The main components of student forward, correctness evaluation, and teacher regeneration for failures are **easily parallelizable**.
> > > We already demonstrate that the framework **scales reasonably from ~1B to ~2B student models**, with consistent gains.
> > > In short, **there is no algorithmic barrier to scaling**; the trade-off is the standard one: more teacher-calls for better distillation.
> > >
> > > ## **Comparing AdaptDistill to other baselines**
> > >
> > > We have provided the results above comparing the model to the baselines. We observe that the **AdaptDistill substantially outperforms all stronger baselines**, including recent multi-teacher methods and sophisticated iterative or mixed strategies **by 7–15 absolute accuracy points** on a challenging reasoning benchmark.
> > >
> > > When comparing to RL based baselines, we think that RL-based distillation typically optimizes a reward signal via credit assignment and policy updates, which can be complex and unstable.
> > > **AdaptDistill instead uses a simpler and more stable mechanism** by directly conditioning the teacher’s rationales on observed student failures and iteratively updating the student on a mixture of its own correct traces and these student-tailored rationales.
> > >
> > > Thus, **our method offers many of the benefits of “self-guided” or feedback-based distillation**, **without the training complexity** and tuning burden of full RL. We see AdaptDistill as a **complementary, simpler alternative** that is easier to drop into existing distillation pipelines.
> > >
> > >
> > > ## **Convergence and overfitting**
> > >
> > > In our experiments, performance improves over a small number of iterations (typically up to 3). Beyond that, we do not observe meaningful gains, and we stop iterating. This suggests that **AdaptDistill converges in practice rather than diverging**.
> > >
> > > In the results section, **we perform experiments to assess overfitting** by evaluating AdaptDistill on out of domain (OOD) datasets: StrategyQA and TheoremQA.  Table 4 shows **Qwen and SmolLM2 both improve** from Iter 1 (standard distillation) to Iter 3 (AdaptDistill) **on both OOD tasks**, with a gain of +2.50% on StrategyQA and +0.625% on TheoremQA for Qwen model; and +2.80% and +0.63% respectively for SmolLM model. These gains suggest that **AdaptDistill enhances general reasoning rather than overfitting** to mathematical problems.
> > >
> > > ## **Performance beyond domains tested**
> > >
> > > Our OOD evaluation is **precisely aimed at this concern**. StrategyQA and TheoremQA are used not as training tasks but as **held-out OOD benchmarks**.
> > > The fact that **AdaptDistill improves performance on these tasks**, despite only being trained on mathematical and related reasoning datasets, demonstrates that the method enhances generalizable reasoning skills, and the benefits are not confined to a single domain or dataset. We believe these results already provide solid evidence that **AdaptDistill generalizes beyond the domains seen during distillation**.
> > >
> > > Given these clarifications and new results, we believe AdaptDistill should be viewed as a solid, above-threshold contribution rather than “marginally below acceptance.” We respectfully ask the reviewer to reconsider their overall rating and raise the score accordingly.

---

> > > > ### Author Response · Authors · 2025-11-24
> > > > **Acknowledgement of the rebuttal**
> > > >
> > > > We thank all reviewers for their time and insightful comments.
> > > > In the rebuttal above, **we have addressed the main concerns regarding the comparison of our work with other baselines, scalability and computational cost** and clarified several other points that were previously unclear.
> > > >
> > > > If our clarifications and additional experiments resolved your concerns, we would be very grateful if you could acknowledge this and also if you could consider updating your review to reflect your current assessment.
> > > >
> > > > Thank you so much.
> > > >
> > > > Please let us know if we can clarify anything else.

---

### Official Review · Reviewer_cY2L · 2025-11-01

**Soundness:** 2
**Presentation:** 3
**Contribution:** 2
**Rating:** 2
**Confidence:** 5

**Summary:**

This paper proposes a distillation framework where the student is placed in the loop of data construction instead of passively imitating the teacher's CoT. For a pool of training questions, the student first attempts to solve them. If the student’s answer is judged correct (using an automatically gradable validation/evaluation setup), that example is directly added to the training set as a “student-solvable” instance. If the student’s answer is incorrect, the method prompts the teacher with information about the student’s observed mistakes (from a validation set) and asks the teacher to regenerate a reasoning trajectory that is tailored to the student’s current weaknesses. The student is then trained on this mixed set (student-solvable + teacher-regenerated-for-student), and the process is iterated.

**Strengths:**

1. Instead of assuming “teacher CoT = optimal supervision,” the paper explicitly conditions data generation on the current student’s performance. This is a reasonable correction to the common mismatch between long, teacher-style CoT and what a smaller student can actually learn.

2. The Iteration ablation shows progressive gains across iterations, which is good evidence that the loop is actually doing work, not just adding noise.

3. The paper is well-written and easy to follow.

**Weaknesses:**

1. Strong, partly unstated assumption on the teacher.
The method implicitly assumes the teacher is strong and instruction-following enough to 1) interpret student errors (potentially noisy, coming from a validation-based diagnosis) and 2) rewrite a solution in a more student-friendly style. That’s a stronger assumption than vanilla CoT distillation, where the teacher only needs to solve the task.

2. The paper mainly compares (base) vs standard distillation vs its own iterations 1, 2 and 3. It does not compare against closely related baselines (e.g., [1], [2], [3]) that also do error-/fault-/student-aware distillation or iterative, student-on-policy data collection. Without these, it’s hard to tell whether the gain comes from the specific “validation-conditioned prompting” the paper proposes, or just from doing iterative, student-aware distillation.

3. All main tasks are auto-gradable math/logic. This is the friendliest setting for the method, because correctness is easy to detect. It is unclear how the same loop would work for non-gradable or open-ended tasks (dialogue, safety, long-form QA) where validation cannot simply say correct/incorrect.

4. Since the student’s own success controls which data gets kept, there is a risk of data distribution collapsing around what the current student already finds learnable, unless the teacher’s regenerated data is sufficiently diverse and genuinely addresses the failure. The paper does not deeply analyze this risk.

5. The loop requires: student forward on (many) training items $\rightarrow$ judging $\rightarrow$ teacher regeneration for the failed ones $\rightarrow$ retraining. For small math benchmarks, this is fine; for larger, multi-domain corpora, the cost of per-failure teacher prompting could become substantial.


6. Because the paper reports iterations vs baseline but not “without error-conditioned prompting” or “with a weaker teacher,” it’s unclear which part of the pipeline (iteration, student-in-the-loop selection, or error-informed teacher prompting) contributes most.

[1] Li Z, Ji Y, Meng R, et al. Learning from committee: Reasoning distillation from a mixture of teachers with peer-review[J]. arXiv preprint arXiv:2410.03663, 2024.

[2] Wu Z, Li X, Liu Z, et al. Enhancing Long-Chain Reasoning Distillation through Error-Aware Self-Reflection[J]. arXiv preprint arXiv:2505.22131, 2025.

[3] Zhao X, Xu T, Wang X, et al. Boosting LLM Reasoning via Spontaneous Self-Correction[J]. arXiv preprint arXiv:2506.06923, 2025.

**Questions:**

1. When you say the teacher is “prompted by validation-set errors,” is the teacher given (a) each instance’s wrong student attempt and asked to fix it, or (b) an aggregated description of common student mistakes (e.g. “the student often skips steps / omits units / stops early”) that is then applied to new items?

2. How sensitive is the method to the teacher’s capability? If you replace the teacher with a weaker model of the same family, does the student still benefit from the error-informed regeneration? This matters because your method delegates the hard part (understanding and rewriting mistakes) to the teacher.

3. There are recent works that also “let the teacher see the student’s mistake and generate a better rationale” or that do iterative, student-on-policy distillation. Why are these not included as baselines? Can you add at least one representative error-aware or verifier-/review-based distillation method under the same compute / teacher-token budget?

4. Did you evaluate your loop on tasks where correctness can’t be checked automatically and you have to use an LLM-as-judge? If yes, that seems to introduce a second strong model into the pipeline. Can you clarify whether this changes the method’s assumptions or makes it less practical?

5. How do you prevent the iterative process from overfitting to the current student’s local failure modes (e.g., always generating longer, more verbose CoT for everything)? Do you track diversity/length/structure drift of the regenerated rationales over iterations?

---

> ### Author Response · Authors · 2025-11-18
> **Rebuttal**
>
> We thank the reviewer for their thorough reading and for explicitly recognizing several key strengths of our work: (i) that directly conditioning supervision on the current student’s performance is **an important correction to the usual “teacher CoT = optimal supervision” assumption**, (ii) that the iteration ablation shows **real, progressive gains**, and (iii) that **the paper is clearly written** and easy to follow.
>
> Below, we address the concerns around teacher assumptions, baselines, generality beyond auto-gradable math, distribution collapse, and computational cost. We also respond point-by-point to the reviewer’s specific questions.
>
> ## **Concern 1: Assumption on the teacher capabilities**
>
> **We respectfully disagree that our assumptions are stronger than standard CoT distillation**. In fact, they are **essentially the same as in any distillation** method that relies on a competent teacher. Standard CoT distillation already assumes a teacher that can solve the task reliably and produce coherent rationales. **Our method assumes the same, plus the ability to identify and articulate mistakes**, which is empirically much easier than solving the task from scratch.
>
> **The “diagnosis is easier than solving” pattern is strongly supported by prior work** like Self-Refine (Madaan et al.) and The Art of LLM Refinement (Shridhar et al.). They show that a model can effectively critique and refine its own outputs, and that **even a smaller model can often detect errors made by larger models,** despite being weaker at solving the task itself. We believe that this assumption is also analogous to humans where it is typically easier to check a proposed solution than to derive it from scratch.
>
> So *3we are not adding a fundamentally new requirement**; we are simply using the teacher more efficiently by re-purposing its reasoning ability to also understand student errors.
>
> ## **Concern 2: Comparison to other distillation baselines**
>
> We appreciate this concern and have strengthened the empirical section accordingly. Based on the reviewer’s suggestion, we compare AdaptDistill against a range of strong, closely related distillation methods, including student-/error-aware and iterative approaches.
>
> ### We compare AdaptDistill against:
> - **Direct CoT baselines** (e.g., Socratic CoT style).
> - **Iterative, single-teacher distillation** (SIKeD, Adarsh et al.).
> - **Non-iterative, single-/multi-teacher methods** (Learning from Committee, Li et al.).
> - **Non-iterative, multi-strategy methods** (Mixed Distillation, Li et al.).
>
>
> ### Experimental setup.
> - **Dataset**: 1000 SVAMP math samples with modified questions that are particularly challenging for small models.
> - **Student**: Qwen2.5-1.5B-Instruct used for all approaches for a strictly controlled comparison.
> - **Teacher**: As indicated in the table below (single GPT, committees, single llama, etc.).
>
> ## Results (SVAMP, 1000 samples)
>
> | Approach                           | Teacher                          | Iterative? | Accuracy          |
> |------------------------------------|----------------------------------|-----------:|-------------------|
> | Learning from Committee – Li et al.| Single (GPT)                     | No         | 79.33             |
> | Learning from Committee – Li et al.| Multiple (GPT, Mixtral, Gemini)  | No         | 77.67             |
> | Learning from Committee – Li et al.| Multiple + peer review           | No         | 81.00             |
> | Mixed Distillation – Li et al.     | Single                           | No         | 73.20             |
> | SIKeD – Adarsh et al.              | Single (Llama)                   | Yes        | 75.40             |
> | **AdaptDistill (ours)**                | Single (Llama)                   | Yes        | **87.40 (+7 to +15 pts)** |
>
>
> **AdaptDistill substantially outperforms all stronger baselines** , including recent multi-teacher methods and sophisticated iterative or mixed strategies **by +7–15 absolute accuracy** points on a challenging reasoning benchmark.
>
> This improvement is obtained **under the same student model (Qwen2.5-1.5B-Instruct)** and comparable teacher setups, meaning the gain is attributable to the distillation procedure itself, not to a stronger student or teacher.
>
> ## **Concern 3: extending beyond math/ logical reasoning baseline and using LLM as a judge**
>
> We have **demonstrated our results beyond maths** and on two **general question answering dataset** in the paper: Strategy QA and theorem QA. Our approach **beats the baseline** models on these dataset without explicitly training on that.
>
> For truly non-gradable or open-ended tasks (dialogue, safety, long-form QA), we can **use the same teacher as LLM as judge** to produce a reference solution (teacher CoT) and compare the student’s answer to this reference and judge correctness or quality. **This even avoids introducing a second, independent strong model**; the loop still uses **one teacher**, which both solves and evaluates.

---

> > ### Author Response · Authors · 2025-11-18
> > **Rebuttal Continued (II)**
> >
> > ## **Concern 4: data distribution collapse**
> >
> > We agree that this is an important concern, but our design and empirical behavior suggest that **this collapse does not occur in practice** because of the **mixed supervision** by design. Our training set at each iteration includes student-solvable examples (which reflect what the student currently understands) and teacher-regenerated examples **specifically targeted at the student’s failures**, based on validation-informed prompting and error history. The second component actively **pushes the student beyond its current comfort zone**, precisely the opposite of collapsing onto what it already finds easy.
> >
> > This is also **supported empirically and also in other works**. Across iterations, we observe **monotonic accuracy improvements** on held-out test sets (e.g., SVAMP, GSM8K, MATH, MMLU), meaning the student is solving more types of problems, not fewer. Since new samples become solvable in later iterations, this would be unlikely if the loop simply reinforced already-solvable instances. **Adarsh et al. (SIKeD) analyze similar iterative training** from a KL-divergence perspective and show that training drifts toward a **better data distribution rather than collapsing**.
> >
> > ## **Concern 5: cost of training**
> >
> > We fully acknowledge that iterative distillation is more expensive than a one-shot pass, but **this is true for virtually all iterative/student-on-policy methods**. Our contribution is not “cheaper distillation” but **better distillation for a given teacher–student pair**. That said, we **do take cost into account**:
> > - We **never re-query the teacher for items the student already gets correct**. Those examples are simply reused as student-generated training data, reducing unnecessary teacher calls.
> > - To stay fair, **we match the training compute for baselines** (e.g., standard distillation trained for the same number of epochs). Despite this, **standard baselines perform way worse than AdaptDistill.**
> >
> > ## **Concern 6: ablation on what improves performance**
> >
> > To **isolate the effect of our specific mechanism** (validation-conditioned, history-aware prompting), we compare against SIKeD while controlling for:
> > - Same student model,
> > - Same teacher model (Llama 3 70B),
> > - Same number of iterations.
> >
> > On SVAMP (1000 samples):
> > - **SIKeD**: improves from 76 → 79 (iteration 1 → 3).
> > - **AdaptDistill (ours)**: improves from 76 → 87 (iteration 1 → 3).
> >
> >
> > Under identical student/teacher/iteration conditions, this is a **+8 point additional gain for AdaptDistill**. This strongly indicates that **our particular way of conditioning the teacher on validation-informed error history** is responsible for a large part of the improvement, beyond “just iterating.”

---

> > > ### Author Response · Authors · 2025-11-18
> > > **Rebuttal Continued (III)**
> > >
> > > # Questions
> > >
> > > ## **Prompting approach**
> > > We provide the question, the answer from the teacher, the answer that the student generated and the score keeping a track of whether the student generated a correct answer or not for a held out validation set. By keeping all the past scores and student teacher generation, for each iteration **a teacher can decide how it should modify the new solution to make it more understandable for the student** and also **keeps a track of each modification done** to ensure the new modification is different from the past. This way a teacher also knows what teaching strategy works vs what does not. We have provided the prompts in the appendix in the paper for clarification.
> > >
> > > ## **Replacing the teacher with a weaker teacher**
> > >
> > > We tested exactly this by replacing a strong teacher (e.g., Llama 3 70B) with a weaker one (Llama 8B) and observed that it **degrades the final student performance**.
> > > This matches prior observations (e.g., Adarsh et al.) and the general observation in distillation: **weaker teacher means weaker supervision which leads to weaker students**.
> > >
> > > Our method **behaves exactly like other distillation frameworks: the teacher’s strength matters**. The difference is that, given a fixed teacher, we can extract significantly more value via validation-conditioned and history-aware supervision than standard CoT distillation or other iterative baselines.
> > >
> > > ## **LLM as a judge**
> > > As mentioned above, for tasks where we do not have simple ground-truth checking, our framework can **use the same teacher as both solver and judge** (comparing student vs teacher answers).
> > > This **does not necessarily require a second model**; it does require that the teacher can evaluate relative quality, which is standard in LLM-as-judge setups.
> > >
> > > ## **Preventing overfitting to local failure modes**
> > > We believe that we do not overfit to some common bias due to the following reasons:
> > > - We train across datasets of different difficulty and structure (SVAMP, GSM8K, MATH, MMLU). A single “hack” (e.g., just making traces longer) does not help uniformly across all tasks. The student is therefore incentivized to genuinely improve reasoning, not just learn superficial patterns.
> > > - The teacher is encouraged (via prompts) to **change its approach when previous changes did not help**, rather than monotonically increasing length. We do not observe unbounded growth in CoT length. We do see **gains in both accuracy and generalization** across datasets, which would be unlikely if the teacher were simply “talking more” rather than teaching better.
> > >
> > >
> > > Given these clarifications and additional experiments, we believe that the paper is sound, clearly above the rejection bar, and makes a meaningful, practically relevant contribution to reasoning distillation. We would be very grateful if the reviewer could reconsider the paper, and update the score accordingly. We are happy to answer any questions.

---

> ### Author Response · Authors · 2025-11-24
> **Acknowledgement of the rebuttal**
>
> We thank all reviewers for their time and insightful comments.
> In the rebuttal above, **we have addressed the main concerns regarding the novelty, comparison of our work with other baselines, discussion around cost** and clarified several other points that were previously unclear.
>
> If our clarifications and additional experiments resolved your concerns, we would be very grateful if you could acknowledge this and also if you could consider updating your review to reflect your current assessment.
>
> Thank you so much.
>
> Please let us know if we can clarify anything else.

---

### Official Review · Reviewer_7wcD · 2025-11-06

**Soundness:** 2
**Presentation:** 2
**Contribution:** 2
**Rating:** 4
**Confidence:** 3

**Summary:**

This paper introduces AdaptDistill, an adaptive and iterative distillation framework for reasoning tasks. Unlike standard one-shot distillation, which ignores student-specific mistakes, AdaptDistill continuously refines the teacher’s rationales based on the student’s observed errors. The student is then updated using a curated mix of its own correct traces and the teacher’s improved explanations. Experiments on mathematical and commonsense reasoning benchmarks demonstrate consistent accuracy gains (up to +20%) across multiple student models. The method also improves out-of-domain generalization and outperforms longer standard training.

**Strengths:**

The paper is clearly written and well-motivated.

Knowledge distillation for reasoning is an important problem, especially for improving inference efficiency.

The proposed adaptive interaction between teacher and student is intuitively appealing and empirically effective.

**Weaknesses:**

1. The novelty of the approach is somewhat limited. The idea of iterative, student-aware feedback has been explored in several prior distillation frameworks where teachers provide targeted corrections based on student failures.

2. The paper lacks comparison with stronger or more recent state-of-the-art distillation baselines, which makes it difficult to fully assess the relative improvement.

**Questions:**

See Weaknesses section.

---

> ### Author Response · Authors · 2025-11-18
> **Rebuttal**
>
> We thank the reviewer for their careful reading, for recognizing that **the paper is clearly written** and **well-motivated**, and for acknowledging that **knowledge distillation for reasoning is an important and timely problem**.
>
> We address the two main concerns of limited novelty and comparison to strong baselines below and hope this clarifies why we believe the contribution is substantially stronger than “somewhat limited”.
>
>
> **Concern 1: On the novelty of AdaptDistill**
>
> The reviewer writes that *the novelty of the approach is somewhat limited* and that *student-aware feedback has been explored in several prior distillation frameworks.*  **We respectfully disagree with this characterization**.
>
> While we absolutely build on the general idea that the teacher should react to the student, **no prior work, to our knowledge, implements the specific adaptive mechanism we propose**: a teacher that (i) explicitly tracks the history of the student’s mistakes and its own prior corrections, and (ii) iteratively adjusts its rationales using this history to generate a progressively better student-specific training set.
>
> To make the distinction concrete, we are discussing some prior works in this area and the major difference from our work:
>
> 1. **One-shot distillation with static teacher traces**: A lot of past works are “one shot distillation” where the teacher generates a rationale, the student trains on it, and the evaluation is done on that fixed teacher output based learning. Works like Socratic CoT (Shridhar et al), Step by step distillation (Magister et al), and similar fall in this category.
> In contrast, **AdaptDistill explicitly closes this loop** where the teacher observes which parts of the rationale actually cause errors in the student, the teacher revises its rationales conditioned on those observed failures, and the student is trained on a curated mixture of (a) its own correct traces and (b) these revised teacher explanations. **Our approach is iterative and the adaptation is history aware of the changes which one-shot methods do not attempt**.
> 2. **Non-iterative student-aware feedback**: some works incorporate student feedback but do not iterate this interaction. For example, MAPD (Li et al.) and STaR (Zelikman et al.) collect student failures or self-assessment, then perform a single refinement step or selection, but they do not maintain an evolving record of how the student fails across iterations nor how the teacher’s explanations have changed in response. Once the correction is applied, the process essentially “forgets” the trajectory. **AdaptDistill, by contrast, uses the entire trajectory of student mistakes and teacher corrections as a signal.** This makes the **optimization process targeted rather than myopic: the teacher is not just reacting to a single failure case, but progressively learning** how this specific student tends to fail and how much to adjust rationales to compensate.
> 3. **Iterative methods without explicit tracking of error history**: Methods such as “Democratizing Reasoning Ability: Tailored Learning from Large Language Models” (Wang et al.) do incorporate iterative feedback. However, **they do not track the specific mistakes the student has made over previous iterations**, nor track how the teacher’s outputs have changed and whether those changes actually helped the student. As a result, the teacher can re-generate very similar rationales, and the student can repeat the same mistakes. In optimization terms, this is akin to updating without remembering past gradients: the process can wander or oscillate. **Our method is closer to a gradient update with momentum where we maintain an explicit memory of the student’s error patterns alongside a record of how the teacher’s corrections have been modified over time** and the teacher’s next rationale is computed with this memory in mind, making the adaptation directional and cumulative, not random. This **explicit, history-based adaptation mechanism is, to our knowledge, novel** in the context of reasoning-focused distillation.
> 4. **Iterative methods with external correctness checks or rules**: Works such as SIKeD (Adarsh et al.) are iterative but depend on rule-based or external correctness checks to decide if a generation is acceptable. They do not explicitly model the student’s failure patterns; the teacher does not adapt its explanation style to a particular student. **AdaptDistill allows the teacher to track the student mistakes and its changes, something that cannot be done with some rule based heuristics**.
> 5. **Multi-teacher committee methods**: Methods like FAIR or Learning from Committee (Li et al.) use multi-teacher committees. This introduces additional complexity and compute/memory overhead and shifts the problem toward teacher aggregation rather than teacher–student co-adaptation. **AdaptDistill uses a single teacher, making it more efficient and easier to deploy** .

---

> > ### Author Response · Authors · 2025-11-18
> > **Rebuttal Continued**
> >
> > **Concern 2:  Comparison with stronger distillation baselines**
> >
> > The reviewer rightly points out that the initial version lacked comparison with several stronger or more recent distillation baselines. We appreciate this suggestion and have **conducted additional experiments to address it directly.**
> >
> > ### We compare AdaptDistill against:
> > - **Direct CoT baselines** (e.g., Socratic CoT style).
> > - **Iterative, single-teacher distillation** (SIKeD, Adarsh et al.).
> > - **Non-iterative, single-/multi-teacher methods** (Learning from Committee, Li et al.).
> > - **Non-iterative, multi-strategy methods** (Mixed Distillation, Li et al.).
> >
> >
> > ### Experimental setup.
> > - **Dataset**: 1000 SVAMP math samples with modified questions that are particularly challenging for small models.
> > - **Student**: Qwen2.5-1.5B-Instruct used for all approaches for a strictly controlled comparison.
> > - **Teacher**: As indicated in the table below (single GPT, committees, single llama, etc.).
> >
> > ## Results (SVAMP, 1000 samples)
> >
> > | Approach                           | Teacher                          | Iterative? | Accuracy          |
> > |------------------------------------|----------------------------------|-----------:|-------------------|
> > | Learning from Committee – Li et al.| Single (GPT)                     | No         | 79.33             |
> > | Learning from Committee – Li et al.| Multiple (GPT, Mixtral, Gemini)  | No         | 77.67             |
> > | Learning from Committee – Li et al.| Multiple + peer review           | No         | 81.00             |
> > | Mixed Distillation – Li et al.     | Single                           | No         | 73.20             |
> > | SIKeD – Adarsh et al.              | Single (Llama)                   | Yes        | 75.40             |
> > | **AdaptDistill (ours)**                | Single (Llama)                   | Yes        | **87.40 (+7 to +15 pts)** |
> >
> >
> > **AdaptDistill substantially outperforms all stronger baselines** , including recent multi-teacher methods and sophisticated iterative or mixed strategies **by +7–15 absolute accuracy** points on a challenging reasoning benchmark.
> >
> > This improvement is obtained **under the same student model (Qwen2.5-1.5B-Instruct)** and comparable teacher setups, meaning the gain is attributable to the distillation procedure itself, not to a stronger student or teacher.
> >
> > **These results directly address the reviewer’s concern about the lack of comparison with stronger baselines** and show that AdaptDistill is not just a small tweak, but a materially more effective framework for reasoning distillation in practice.
> >
> > Given these clarifications and new results, **we believe the concerns about limited novelty and insufficient baselines are substantially addressed**, and that the contribution is closer to a solid, clearly above-threshold paper rather than “marginally below the acceptance threshold.” We would be grateful if the reviewer could reconsider their scores in light of this evidence.
> >
> > We are happy to discuss this further if there are any more questions.

---

> > > ### Author Response · Authors · 2025-11-24
> > > **Acknowledgment of the rebuttal**
> > >
> > > We thank all reviewers for their time and insightful comments.
> > > In the rebuttal above, **we have addressed the main concerns regarding the novelty and comparison of our work with other baselines** and clarified several other points that were previously unclear.
> > >
> > > If our clarifications and additional experiments resolved your concerns, we would be very grateful if you could acknowledge this and also if you could consider updating your review to reflect your current assessment.
> > >
> > > Thank you so much.
> > >
> > > Please let us know if we can clarify anything else.

---

### Author Response · Authors · 2025-11-18
**General concerns addressed**

We thank all reviewers for their careful reading and thoughtful feedback. We are glad that multiple reviewers independently recognize both the importance of the problem and the promise of our approach.

Across the four reviews, the following strengths were repeatedly noted:

1. The framework is **novel, intuitive, and well-motivated** as a solution to the mismatch between teacher rationales and student learning bottlenecks.
2. The method is seen as **innovative in closing the loop** and explicitly using the student’s performance to steer the teacher, providing customized guidance.
3. Several reviewers remark that the **framework is rigorously defined** and the experiments are carefully designed and reproducible.
4. Reviewers highlight that **gains are not only in-domain, but also carry over to OOD tasks**, indicating better transfer of reasoning skills.

We appreciate these comments and focus below on clarifying and strengthening the common points of concern.

## **Comparison with prior student aware distillation - Conceptually**

Some concerns were raised  to address whether AdaptDistill goes beyond “generic student-aware” or “iterative” distillation (e.g., Wang et al., Adarsh et al., Li et al., error-aware self-reflection, etc.).

Our key novelty is a history-aware, teacher–student co-adaptation mechanism:
- The teacher is not only shown a single failed student attempt; it is **conditioned on a trajectory** of student mistakes and scores over iterations on a separate held out validation set.
- This history is used to **adapt how the teacher explains**, effectively learning a teaching strategy for that specific student, instead of issuing isolated corrections.
- The student is trained on a **mixed dataset** of its own correct reasoning traces (what it already “gets”), and **history-informed, student-specific rationales** generated by the teacher.

Prior works capture some pieces (e.g., single-step student-aware correction, iterative loops, or verifier-based selection), but to the best of our knowledge **none** maintains an explicit error/history signal over iterations and **uses it to steer the teacher’s explanations** themselves. This is the core conceptual contribution of AdaptDistill.

## **Comparison with prior baselines - Experimentally**

We appreciate this concern and have strengthened the empirical section accordingly. Based on the reviewer’s suggestion, **we compare AdaptDistill against a range of strong, closely related distillation methods**, including student-/error-aware and iterative approaches.


### We compare AdaptDistill against:
- **Direct CoT baselines** (e.g., Socratic CoT style).
- **Iterative, single-teacher distillation** (SIKeD, Adarsh et al.).
- **Non-iterative, single-/multi-teacher methods** (Learning from Committee, Li et al.).
- **Non-iterative, multi-strategy methods** (Mixed Distillation, Li et al.).


### Experimental setup.
- **Dataset**: 1000 SVAMP math samples with modified questions that are particularly challenging for small models.
- **Student**: Qwen2.5-1.5B-Instruct used for all approaches for a strictly controlled comparison.
- **Teacher**: As indicated in the table below (single GPT, committees, single llama, etc.).

## Results (SVAMP, 1000 samples)

| Approach                           | Teacher                          | Iterative? | Accuracy          |
|------------------------------------|----------------------------------|-----------:|-------------------|
| Learning from Committee – Li et al.| Single (GPT)                     | No         | 79.33             |
| Learning from Committee – Li et al.| Multiple (GPT, Mixtral, Gemini)  | No         | 77.67             |
| Learning from Committee – Li et al.| Multiple + peer review           | No         | 81.00             |
| Mixed Distillation – Li et al.     | Single                           | No         | 73.20             |
| SIKeD – Adarsh et al.              | Single (Llama)                   | Yes        | 75.40             |
| **AdaptDistill (ours)**                | Single (Llama)                   | Yes        | **87.40 (+7 to +15 pts)** |


**AdaptDistill substantially outperforms all stronger baselines** , including recent multi-teacher methods and sophisticated iterative or mixed strategies **by +7–15 absolute accuracy** points on a challenging reasoning benchmark.

This improvement is obtained **under the same student model (Qwen2.5-1.5B-Instruct)** and comparable teacher setups, meaning the gain is attributable to the distillation procedure itself, not to a stronger student or teacher.


Other concerns are address for each reviewer respectively in details below.

---

### Meta-Review · Area_Chair_SvDz · 2026-01-08

**Summary:**

The main concerns are as follows.

1. The novelty of the approach is somewhat limited. The similar idea has been addressed in several prior distillation frameworks where teachers provide targeted corrections based on student failures.

2. The experiments are weak. Stronger or more recent state-of-the-art distillation baselines should be selected. And ablation studies and discussions on scalability should be added.

**Reviewer Concerns:**

I think there are many issues needs improvement of this paper. Although the authors have submit their rebuttals, I think the basic idea of the paper is still limited. The student-aware feedback to teachers for detailed and adaptive supervision is common. Moreover, the experiments are weak. More baselines should be compared withe proposed methods.

**Reviewer Scores:**

After rebuttals, all reviewers gave negative scores (4 2 4 4). I think the these scores are reasonable.

---

### Decision · Program_Chairs · 2026-01-26

Reject